# Mechanical Compression of Human Airway Epithelial Cells Induces Release of Extracellular Vesicles Containing Tenascin C

**DOI:** 10.3390/cells11020256

**Published:** 2022-01-13

**Authors:** Chimwemwe Mwase, Thien-Khoi N. Phung, Michael J. O’Sullivan, Jennifer A. Mitchel, Margherita De Marzio, Ayşe Kılıç, Scott T. Weiss, Jeffrey J. Fredberg, Jin-Ah Park

**Affiliations:** 1Department of Environmental Health, Harvard T.H. Chan School of Public Health, Boston, MA 02115, USA; chimwemwemwase@g.harvard.edu (C.M.); tkphung@hsph.harvard.edu (T.-K.N.P.); michael.osullivan@mail.mcgill.ca (M.J.O.); mitchel@hsph.harvard.edu (J.A.M.); nhmdm@channing.harvard.edu (M.D.M.); restw@channing.harvard.edu (S.T.W.); jfredber@hsph.harvard.edu (J.J.F.); 2Channing Division of Network Medicine, Department of Medicine, Brigham and Women’s Hospital and Harvard Medical School, Boston, MA 02115, USA; ayse.kilic@channing.harvard.edu

**Keywords:** asthma, airway remodeling, bronchospasm, mechanical compression, airway epithelial cells, extracellular matrix, tenascin C, extracellular vesicles

## Abstract

Aberrant remodeling of the asthmatic airway is not well understood but is thought to be attributable in part to mechanical compression of airway epithelial cells. Here, we examine compression-induced expression and secretion of the extracellular matrix protein tenascin C (TNC) from well-differentiated primary human bronchial epithelial (HBE) cells grown in an air–liquid interface culture. We measured *TNC* mRNA expression using RT-qPCR and secreted TNC protein using Western blotting and ELISA. To determine intracellular signaling pathways, we used specific inhibitors for either ERK or TGF-β receptor, and to assess the release of extracellular vesicles (EVs) we used a commercially available kit and Western blotting. At baseline, secreted TNC protein was significantly higher in asthmatic compared to non-asthmatic cells. In response to mechanical compression, both *TNC* mRNA expression and secreted TNC protein was significantly increased in both non-asthmatic and asthmatic cells. TNC production depended on both the ERK and TGF-β receptor pathways. Moreover, mechanically compressed HBE cells released EVs that contain TNC. These data reveal a novel mechanism by which mechanical compression, as is caused by bronchospasm, is sufficient to induce the production of ECM protein in the airway and potentially contribute to airway remodeling.

## 1. Introduction

A hallmark of asthma is aberrant airway remodeling. While the origin of airway remodeling is not well understood, a growing body of evidence suggests that airway epithelial cells are a causal factor [1,2,3,4]. In particular, during asthma exacerbations, airway narrowing causes mechanical compression of airway epithelial cells, which then produce pathologic mediators thereby contributing to airway remodeling [5,6,7]. Mechanical compression applied to well-differentiated human bronchial epithelial (HBE) cells activates multiple signaling cascades, including epidermal growth factor receptor (EGFR), protein kinase C (PKC), extracellular signal-regulated kinase (ERK), and transforming growth factor-β (TGF-β) receptor, all of which are linked to a variety of pathophysiologic features of airway remodeling and asthma [8,9,10,11,12,13,14,15,16]. Furthermore, our recent RNA sequencing analysis revealed that mechanical compression of non-asthmatic HBE cells induces transcriptional changes that recapitulate an asthmatic phenotype, including the upregulation of type 2 inflammatory genes, epithelial repair genes, and extracellular matrix (ECM) remodeling genes [8]. In this study, we focus on the role of mechanical compression of human bronchial epithelial cells in ECM production, with a particular focus on tenascin C (TNC).

TNC is an ECM glycoprotein that modulates cellular processes such as cell adhesion, proliferation, and migration during embryonic development and tissue repair [17,18]. During embryonic development, TNC expression is abundant in many organs [19,20]. In the developing lung, for example, TNC accumulates at the tips of the growing airway and is required for branching morphogenesis and alveolarization [21,22]. In adult tissues under normal conditions, TNC expression is mostly absent or significantly decreased, and is restricted to tissues bearing high tensile stress and some stem cell niches [17,23]. However, under certain disease conditions or during tissue remodeling, TNC expression is rapidly increased [17,23,24,25,26,27,28]. For example, in patients with asthma, abundant TNC expression is detected in the subepithelial basement membrane of the airway at baseline and further increased in response to an allergen challenge [29,30,31,32]. Furthermore, in patients with asthma, serum TNC levels correlate with disease severity, indicating that TNC may serve as a biomarker as well as a pathologic mediator for asthma [33,34]. Further, TNC-deficient mice are protected against allergic inflammation, suggesting that TNC may play a causal role in the progression of asthma [35].

TNC expression is induced by various growth factors, inflammatory cytokines, viral infections, and mechanical stress [8,17,18,36,37,38,39,40,41,42]. In HBE cells, TNC expression is increased by rhinovirus infection, a risk factor for asthma, and type 2 cytokines including IL-4 and IL-13, critical mediators of asthma development [39,40,41]. In our own studies using HBE cells, TNC expression and secretion are increased by mechanical compression, which mimics bronchospasm in asthma [42]. However, the underlying signaling pathways that regulate TNC expression and secretion from airway epithelial cells remain to be elucidated.

While abundant TNC protein is detected in the asthmatic airway, little is known about the source of extracellular TNC or the mechanisms of TNC secretion and its subsequent deposition in the extracellular environment of the airway. In non-epithelial cells, one of the increasingly investigated mechanisms of extracellular TNC secretion is through extracellular vesicles (EVs) [43]. EVs are membrane-bound particles that play a role in intercellular communication by shuttling biological material between cells [44,45,46]. Thus, in the asthmatic airway TNC may be transported from its cellular origin to the extracellular environment through EVs. Here, we demonstrate in HBE cells that EVs containing TNC are released in response to mechanical compression. We establish, further, that compression-induced TNC expression and secretion in HBE cells are regulated by ERK and TGF-β receptor pathways.

## 2. Materials and Methods

### 2.1. Culture of Primary Human Bronchial Epithelial HBE Cells

Primary HBE cells were obtained from the Cystic Fibrosis Center Tissue Procurement and Cell Culture Core, under the approved Protocol (No. 03–1396) by the Institutional Review Board at the University of North Carolina at Chapel Hill. We denoted non-asthmatic cells for the cells isolated from eight donors with no history of smoking or chronic lung disease and asthmatic cells for the cells isolated from three donors with fatal asthma and one donor with non-fatal asthma (Appendix A). As described previously [12,13,47,48,49,50], passage 2 of primary HBE cells were cultured and maintained in air–liquid interface (ALI). For the ALI culture, we used a 1:1 mixture of Dulbecco’s modified Eagle’s medium (DMEM, Lonza, Basel, Switzerland) and Bronchial epithelial cell growth basal medium (BEBM, Lonza) supplemented with Bronchial epithelial growth medium (BEGM) SingleQuot kit supplement and growth factors (Lonza), nystatin (20 units/mL, Sigma Aldrich, St. Louis, MO, USA), retinoic acid (50 nM, Sigma Aldrich), and bovine serum albumin (1.5 µg/mL, Sigma Aldrich). In this study, we used the HBE cells on ALI days 14–16, where we consistently observed well-differentiated epithelial cell phenotypes [11,16,51,52].

### 2.2. Application of Mechanical Compression to HBE Cells

To mimic bronchospasm during an asthma exacerbation, well-differentiated primary HBE cells in ALI culture were subjected to apical-to-basal mechanical compression with a magnitude of 30 cm H_2_O pressure for 3 h, as previously described [6,12,47,50]. Sham control cells received 0 cm H_2_O pressure.

### 2.3. Inhibition of the ERK or TGF-β Receptor Pathway

To determine the cellular signaling pathways by which mechanical compression induced TNC expression and secretion, we used a MEK inhibitor (U0126; 10 µM, Tocris, Bristol, UK) and a TGF-β receptor I inhibitor (SB431542; 10 µM, Tocris). Either inhibitor was added to the basolateral conditioned media of HBE cells at 1 h prior to the application of mechanical compression. As a vehicle control for the inhibitors, we used 0.1% DMSO. Cells, basolateral conditioned media, and apical washes were collected at 3 h and 24 h after the initiation of compression. We harvested and stored 1.5 mL of the basolateral conditioned media for further analyses. To harvest the apical washes, we added 0.25 mL of culture media to the apical surface, incubated for 10 min at 34 °C in a 5% CO_2_ incubator and then collected the apical washes for further analyses.

### 2.4. RNA-Sequencing

Total RNA was isolated from HBE cells (*n* = 6 donors with no history of lung disease) using the RNeasy Mini Kit (Qiagen, Hilden, Germany), according to the manufacturer’s protocol. Bulk RNA sequencing analysis was performed using a NovaSeq instrument (Illumina, San Diego, CA, USA) by The Bauer Core Facility at Harvard University. Reads were mapped to the GRCh38 reference genome using Salmon [53]. Gene expression (pseudo counts from Salmon) was normalized using tximport (v 1.18.0) [54] and DESeq2 (v 1.30.1) [55]. Genes with low counts (less than or equal 10 counts in at least 70 of 72 samples) were filtered out, and differential expression analyses were performed with FDR < 0.05.

### 2.5. RT-qPCR

Using 1 ug of total RNA isolated as described above, we prepared 20 ng of cDNA using MultiScribe reverse transcriptase (Applied Biosystems, Forster City, CA, USA). The cDNA was used to perform real-time RT-qPCR using 2× SYBR Green master mix (Life Technologies, Grand Island, NY, USA) and the primers listed below (Table 1). The fold-change for *TNC* normalized to *GAPDH* was calculated using the delta–delta Ct method [56].

### 2.6. Enzyme-Linked Immunosorbent Assay (ELISA)

We quantified the amount of TNC in the basolateral conditioned media and apical washes using an enzyme-linked immunosorbent assay (ELISA) kit (ab213831; Abcam, Cambridge, UK). The validation of the specificity of the ELISA kit was performed using a recombinant human TNC protein (CC065; Sigma Aldrich).

### 2.7. Western Blot Analysis

As described previously [13,16], using Western blot analysis we detected cellular protein and secreted protein into basolateral conditioned media. Primary antibodies against TNC (33352; Cell Signaling Technology, Danvers, MA, USA), GAPDH (5174S; Cell Signaling Technology), and transferrin (PA3-913; Thermo Fisher Scientific, Waltham, MA, USA) were used. GAPDH and transferrin were detected as loading controls for cellular protein and secreted protein, respectively.

### 2.8. Isolation and Validation of Extracellular Vesicles

After the initiation of mechanical compression, basolateral conditioned media were collected at 24 h. Basolateral conditioned media were centrifuged at 1200× g for 20 min at 4 °C to remove cell debris and to collect cell-free supernatant. To isolate the EV fraction from cell-free supernatant, we used the commercially available kit, MagCapture Exosome Isolation Kit PS (293-77601; Fujifilm WAKO Pure Chemical Corporation, Tokyo, Japan). Following the instruction, we incubated cell-free supernatant for 6 h at 4 °C. Then, using a DynaMag^TM^ 2 (12321D; Invitrogen, Waltham, MA, USA), we separated the magnetic bead-bound fraction as an EV fraction and the remaining supernatant as a non-EV fraction. The bead-bound EVs were boiled with 2× Laemmli sample buffer (1610737EDU; BIO-RAD, Hercules, CA, USA) containing 1 M dithiothreitol at 100 °C for 6 min and used for Western blot analysis. To validate EV fractions by Western blot analysis, we detected known EV markers [13,44,50,57,58,59], including CD9 (D801A; Cell Signaling Technology) and tissue factor (AF2339; R&D Systems, Minneapolis, MN, USA).

### 2.9. Statistical Analysis

For each condition, the results are expressed as mean ± SEM, and statistical analysis was performed using GraphPad Prism 9 software (GraphPad Software Inc., San Diego, CA, USA). In all experiments, a two-way ANOVA with Bonferroni post-hoc test was utilized to analyze the differences between groups. A *p*-value less than 0.05 was considered statistically significant.

## 3. Results

### 3.1. Mechanical Compression Induces TNC mRNA Expression and Secretion in HBE Cells

Using RNA sequence analysis, we previously identified that mechanical compression induces TNC expression in non-asthmatic HBE cells [42]. Here, we compared *TNC* mRNA expression in HBE cells cultured from non-asthmatic and asthmatic donors at baseline and in response to mechanical compression (Figure 1). At baseline, *TNC* mRNA expression was marginally higher in asthmatic cells (1.5-fold, *p* = 0.042) compared to non-asthmatic cells only at the 3 h time point, but not at the 24 h time point (Appendix A). In the non-asthmatic cells, *TNC* mRNA expression was significantly increased by 3.9-fold (*p* < 0.001) at 3 h and 2.9-fold (*p* < 0.01) at 24 h after compression compared to their time-matched controls (Figure 1A). Conversely, in the asthmatic cells, *TNC* mRNA expression was significantly increased by 2.1-fold (*p* < 0.01) at 3 h and 1.4-fold at 24 h after compression (Figure 1B). At either 3 or 24 h, the extent of the increased *TNC* mRNA expression in response to compression did not differ between non-asthmatic and asthmatic cells.

While mechanical compression induced *TNC* mRNA expression in HBE cells, we did not detect increased cellular TNC protein at any time points we examined (at 24, 48, and 72 h post-compression) (Appendix A). To compare the amount of TNC secretion from HBE cells between non-asthmatic and asthmatic cells from four donors each, we measured the level of TNC protein by ELISA in the basolateral conditioned media collected at 24 h after compression (Figure 1C). Similar to our previous report [42], TNC was constitutively secreted to the basolateral side of the well-differentiated HBE cells in ALI culture. At baseline, constitutive TNC secretion was significantly higher in asthmatic HBE cells than in non-asthmatic HBE cells (asthma: 22.3 vs. non-asthma: 5.8 ng/well, *p <* 0.05). In both non-asthmatic and asthmatic HBE cells, mechanical compression induced TNC secretion. While constitutive TNC secretion was higher in asthmatic cells, the amount of compression-induced TNC secretion did not differ between non-asthmatic and asthmatic cells (non-asthma: 56 vs. asthma: 60 ng/well).

### 3.2. ERK and TGF-β Receptor Pathways Regulate the Compression-Induced TNC mRNA Expression and Protein Secretion

To determine intracellular signaling pathways by which TNC is induced by mechanical compression, we first used bulk RNA sequencing analysis to identify mechanical compression-responsive genes that are regulated by either ERK or TGF-β receptor signaling pathways. We assessed differentially expressed (DE) genes at 3 and 24 h post-compression in the presence or absence of either inhibitor for MEK (U0126) or TGF-β receptor (SB431542). We identified *TNC* as one of the compression-responsive DE genes that was downregulated in the presence of either ERK or TGF-β inhibition. At 3 h, treatment with vehicle, U0126, or SB431542 did not affect the expression level of *TNC*. Contrarily, only pretreatment with U0126 (*p* < 0.0001) significantly decreased the mechanical compression-induced *TNC* expression (Figure 2A). At 24 h, we observed that the baseline expression of *TNC* was decreased only by treatment with U0126 (*p* < 0.0001) but not with vehicle or SB431542. However, pretreatment with either U0126 (*p* < 0.0001) or SB431542 (*p* < 0.0001) blocked the mechanical compression-induced *TNC* expression (Figure 2B).

To validate the data from the RNA sequencing analysis, we measured *TNC* expression by RT-qPCR in an independent experiment. Here, in three non-asthmatic donor HBE cells, we determined *TNC* mRNA expression at 24 h post-compression in the presence or absence of either U0126 or SB431542. At baseline, *TNC* mRNA expression was not significantly different in the cells between pretreated with the vehicle and either U0126 or SB431542 (Figure 2C). In response to mechanical compression, *TNC* mRNA expression was increased by 2.3-fold (*p* < 0.05) in the presence of vehicle control, while it was significantly attenuated by 0.97-fold (*p* < 0.01) and 0.93-fold (*p* < 0.01) in the presence of U0126 and SB431542, respectively (Figure 2C). These data demonstrate that pretreatment with either U0126 or SB431542 significantly attenuated the mechanical compression-induced *TNC* mRNA expression (Figure 2C).

In ALI culture, HBE cells are well-differentiated and polarized. In the polarized cells, intracellular signaling pathways may have a preferential effect on the protein trafficking to the basolateral side versus the apical side. Thus, here we measured both basolateral and apical secretions of TNC. First, we detected basolateral secretion of TNC by both Western blot analysis and ELISA (Figure 3A,B). The levels of secreted TNC detected by Western blot analysis and measured by ELISA were comparable. Similar to *TNC* mRNA expression, either inhibitor did not affect the constitutive secretion of basolateral TNC. At baseline, the amount of basolateral TNC was 10 ng/well. In response to mechanical compression, basolateral TNC was increased to 62 ng/well (Figure 3B). In the presence of either inhibitor, compression-induced basolateral TNC was significantly attenuated to 9 ng/well (U0126) and 12 ng/well (SB431542) (Figure 3B). Next, we examined the apical secretion of TNC by ELISA because we validated the specificity of the ELISA assay using basolateral conditioned media. Similar to basolateral secretion, either inhibitor did not affect the constitutive secretion of apical TNC. At baseline, the amount of apical TNC was 3 ng/well. In response to mechanical compression, apical TNC was increased to 17 ng/well (Figure 3C). In the presence of either inhibitor, compression-induced apical TNC was significantly attenuated to 5 ng/well (U0126) and 11 ng/well (SB431542) (Figure 3C). These data indicate that TNC is secreted both apically and basolaterally, but the basolateral secretion of TNC is predominant and could be the major source of extracellular TNC.

### 3.3. Mechanical Compression Induces the Release of TNC Protein Contained in EVs

To determine the mechanism of extracellular TNC secretion from compressed HBE cells, we isolated the EV fraction from basolateral conditioned media. We used Western blot analysis to assess the three fractions obtained during the isolation process: (1) the whole conditioned media, (2) non-EV fraction, and (3) EV fraction. In the whole conditioned media from compressed cells, we detected TNC and the following three proteins: YKL-40, a marker for the non-EV fraction, and CD9 and tissue factor, the markers for the EV fraction [13,44,50,57,58,59] (Figure 4). In the EV fraction, we detected CD9 and tissue factor. In the non-EV fraction, we detected YKL-40. While TNC protein was detected predominantly in the non-EV fraction, it was also prominently detected in the EV fraction suggesting TNC is secreted by multiple mechanisms.

## 4. Discussion

This study examines the role of mechanical compression that is associated with bronchospasm in the modulation of the extracellular environment during the development of asthma. We focused on TNC, an ECM glycoprotein that is highly expressed in asthmatic airways and differentially overexpressed in compressed HBE cells [29,30,31,32,33,34,42]. The mechanical compression that we applied to cells was comparable to the magnitude that is associated with maximal bronchospasm during asthma exacerbations [5,60]. In both non-asthmatic and asthmatic HBE cells, our data demonstrate that mechanical compression increased *TNC* mRNA expression and protein secretion. Constitutive TNC secretion detected at baseline was greater in asthmatic cells compared to non-asthmatic cells, but stimulated TNC secretion in response to mechanical compression was substantially greater than at baseline but not different between asthmatic and non-asthmatic cells. Our data further indicate that *TNC* mRNA expression and protein secretion depended on the activation of ERK or TGF-β receptors. Furthermore, our data reveal that mechanically compressed HBE cells released EVs that contain TNC.

In patients with asthma, abundant TNC expression in the airway is detected in the subepithelial basement membrane and significantly increased after allergen challenge [29,30,31,32]. While the source of this extracellular TNC protein in the airway basement membrane has not previously been elucidated, our data suggest that airway epithelial cells are a major source of this elevated extracellular TNC. Here, we validated that well-differentiated bronchial epithelial cells are a source of extracellular TNC [42] and discovered that TNC is constitutively secreted from HBE cells even in the absence of stimulation (Figure 1C and Figure 3). At baseline, the constitutive secretion of TNC from asthmatic HBE cells was significantly higher than that from non-asthmatic HBE cells (Figure 1C). Similar to the previously reported RNA sequencing analysis [8], we detected higher *TNC* mRNA expression in asthmatic cells at one of the two time points **(**Appendix A). However, due to the small sample size and the high variability among asthmatic cells, this has to be further examined using asthmatic cells that can be associated with the status and history of the disease. In both non-asthmatic and asthmatic HBE cells, mechanical compression caused increased mRNA expression and secreted protein of TNC (Figure 1A–C). However, there was no significant difference in the levels of stimulated TNC production between non-asthmatic and asthmatic HBE cells. These findings indicate that, regardless of the disease state and in the absence of inflammatory cells, bronchospasm alone can induce *TNC* mRNA expression and extracellular TNC production from airway epithelial cells. These findings highlight the important role of bronchospasm in airway physiology beyond asthma because bronchospasm is also experienced in individuals with obesity, with other chronic lung diseases including COPD, or after exposure to environmental pollutants such as particulate matter or ozone [61,62,63,64].

While *TNC* mRNA expression and protein secretion are associated with multiple pathologic conditions [17,18,36,37,38,39,40,41], underlying regulatory mechanisms in HBE cells are yet to be fully elucidated. Among multiple pathologic factors, TGF-β is a potent inducer of TNC expression in various cell types, including fibroblasts [26,38,65]. The TGF-β receptor transduces signals through canonical and non-canonical pathways [66]. The canonical TGF-β pathway is primarily mediated through the SMAD-dependent pathway, while the non-canonical pathway is mediated through several SMAD-independent pathways, including ROCK, JNK, and ERK [66]. In fibroblasts, both SMAD and non-SMAD pathways are involved in the TGF-β-induced TNC expression [38,67]. In HBE cells, TGF-β secretion is increased by mechanical compression and plays an integral role in goblet cell hyperplasia, fibrosis, and subepithelial collagen deposition [9,14,68]. Thus, we hypothesized that in HBE cells TGF-β signaling pathway could regulate mechanical compression-induced TNC production. Pharmacological inhibitors of either ERK or TGF-β receptor I activity significantly attenuated mechanical compression-induced *TNC* mRNA expression (Figure 2), suggesting that the transcriptional regulation of epithelial *TNC* expression depends on ERK or the TGF-β receptor. In addition, inhibition of ERK or TGF-β receptor subsequently attenuated mechanical compression-induced secretion of TNC both basolaterally and apically (Figure 3). While we did not further determine if the attenuated secretion was the result of attenuated *TNC* mRNA expression or an attenuated secretory mechanism, our data clearly indicate that TNC secretion from bronchial epithelial cells depends on either ERK or the TGF-β receptor. Another limitation is that we did not compare apical secretion between non-asthmatic HBE cells and asthmatic HBE cells. Because the apical and basal surfaces of the epithelium interact with different types of immune and resident cells, the function and preference of the secreted TNC to either compartment should be a future direction to investigate. Despite this limitation, our data show that the amount of basal secretion is significantly greater than that of an apical secretion at either baseline or in response to mechanical compression.

EVs play a key role in cell-to-cell communication by transporting various cellular cargo from the donor cells to the recipient cells [44,45,59,69,70]. Thus, EVs were explored as a mechanism of exporting various biological components from the cells to the extracellular environment [43]. Extracellular TNC was detected in EVs released from various cancer cell types [43,71] Clinically, EVs containing TNC were detected in the plasma of patients with cancer [43,72,73]. In patients with COVID-19, EVs isolated from plasma contain abundant TNC compared with that from healthy controls [74]. This study further revealed that these circulating EVs can trigger proinflammatory signals in the cells of a distant organ [74]. In addition to the observations of TNC contained in EVs, EVs are suggested as a required mechanism for the extracellular deposition of TNC by tumor cells and fibroblasts [43,75,76]. In a study using mouse embryonic fibroblasts, the abrogation of EV release correlates with impaired TNC deposition and leads to intracellular accumulation of TNC [75]. In the same study, the delivery of TNC-containing EVs into TNC knockout mice by intravenous injection resulted in the accumulation of TNC in different organs including the liver and lungs, suggesting that EVs contribute significantly to extracellular TNC deposition. Additionally, in the study using bronchial epithelial cell line, BEAS-2B cells, TNC secretion is equally dependent on the non-EV mediated and EV-mediated pathways [41].

Here we examined extracellular TNC secretion from HBE cells in response to mechanical compression. Our previous studies demonstrate that mechanically compressed HBE cells release EVs that contain tissue factor protein [13,50]. The tissue factor was exclusively contained in the EV fraction, and here we used tissue factor as a marker of the EV fraction. In this study, we identified that TNC secreted from compressed HBE cells was contained in both non-EV and EV fractions (Figure 4). This suggests that extracellular TNC secreted from airway epithelial cells depends on both non-EV mediated and EV-mediated pathways in the lung. While abundant TNC is detected in the subepithelial basement membrane, the EV-mediated secretion of TNC potentiates the role of extracellular TNC not only in the lung but also outside of the lung. For example, in the patients with refractory neutrophilic asthma, serum TNC level is significantly higher than that in either the subjects with non-refractory asthma or healthy individuals [34] suggesting that TNC may contribute to persistent asthma symptoms or inflammation in patients with uncontrolled and severe asthma. The cellular cargo contained in EVs can remain stable for a prolonged period of time and be transported to cells in distant organs [44,45,59,69,70]. This is the first study to show that a non-inflammatory cue can induce the release of EVs that contain an ECM protein, TNC. To identify additional proteins contained in epithelial cell-derived EVs that may play critical roles in ECM deposition in asthma, proteomic analysis on the EVs is required.

In summary, our data provide a causal link between bronchospasm and extracellular TNC production from airway epithelial cells. We showed that asthmatic HBE cells constitutively secrete greater amounts of TNC compared to non-asthmatic HBE cells, suggesting that epithelial cell-derived TNC could be a major source of extracellular TNC deposited in the airway of patients with asthma. TNC secretion from HBE cells is substantially induced by mechanical compression in the absence of inflammatory cells, suggesting that bronchospasm alone can further augment the extracellular TNC production during asthma exacerbations. Importantly, the role of bronchospasm in TNC secretion may not be limited to patients with asthma because the cells from donors with no history of lung diseases also showed increased secretion of TNC by mechanical compression. These data indicate that TNC can be increasingly secreted from individuals who are prone to experience bronchospasm due to underlying conditions such as obesity or recurrent exposures to environmental pollutants. Finally, TNC contained in EVs that are released from compressed HBE cells could be transported to distal tissues. Our data reveal a novel mechanism by which bronchospasm alone is sufficient to induce the deposition of extracellular matrix protein in the airway, thereby contributing to remodeling.

## Figures and Tables

**Figure 1 cells-11-00256-f001:**
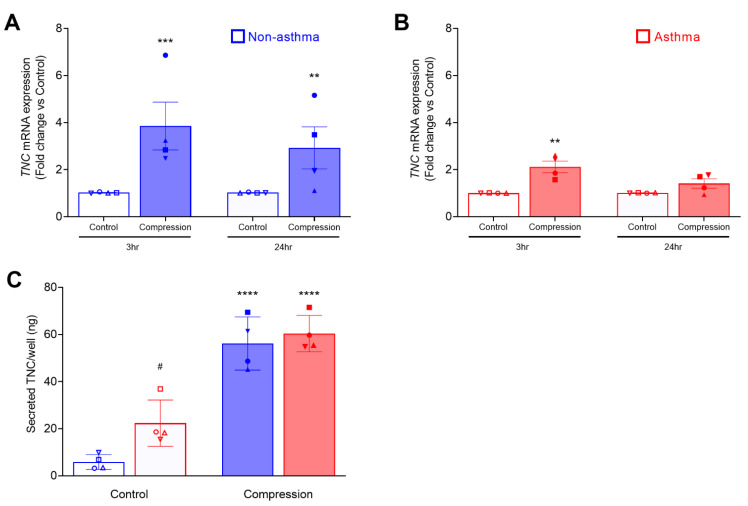
Mechanical compression induces mRNA expression and secretion of TNC from HBE cells. *TNC* mRNA expression was measured at 3 h and 24 h post-compression of non-asthmatic cells (mean ± SEM, 4 donors) (**A**) and asthmatic cells (mean ± SEM, 4 donors) (**B**). *TNC* mRNA expression was normalized to *GAPDH*. The amount/well of TNC secreted in the basolateral conditioned media was measured by ELISA (mean ± SEM, 4 non-asthma and 4 asthma donors) (**C**). *** p* < 0.01, *** *p* < 0.001, **** *p* < 0.0001, significantly different from time-matched no pressure control, ^#^ *p* < 0.05 significantly different between non-asthma and asthma, analyzed by two-way ANOVA with Bonferroni’s post-hoc test. Each symbol represents each of the four donors. The open symbols represent control condition, and the closed symbols represent compressed condition in both non-asthmatic (blue symbols; **A**,**C**) and asthmatic HBE cells (red symbols; **B**,**C**).

**Figure 2 cells-11-00256-f002:**
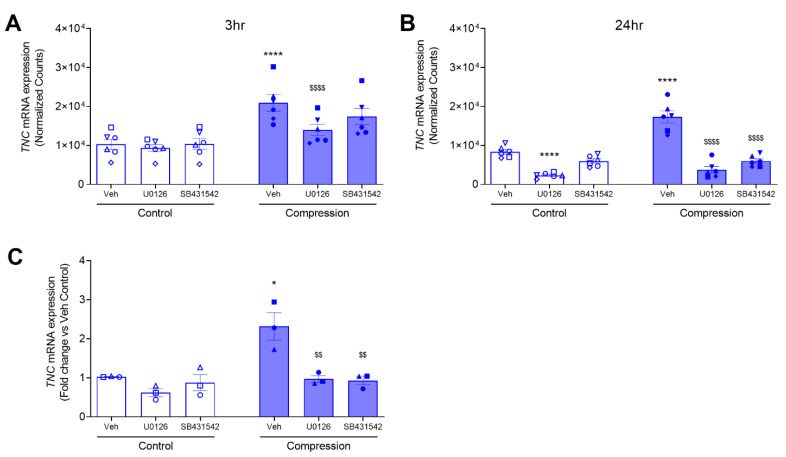
Compression-induced *TNC* mRNA expression depends on TGF-β receptor and ERK pathways. Expression plots showing normalized gene expression levels of *TNC* at 3 h (**A**) and 24 h (**B**) in compressed HBE cells in the absence or presence of either U0126 or SB431542 (mean ± SD, 6 non-asthma donors). *TNC* mRNA expression was validated by qRT-PCR and normalized to *GAPDH* (mean ± SEM, 3 non-asthma donors) (**C**). ** p* < 0.05, **** *p* < 0.0001, significantly different from time-matched no pressure control, ^$$^ *p* < 0.01, ^$$$$^ *p* < 0.0001, significantly different from vehicle with pressure, analyzed by two-way ANOVA with Bonferroni’s post-hoc test. Each symbol represents each of the six (**A**,**B**) or four (**C**) donors. The open symbols represent control condition, and the closed symbols represent compressed condition in non-asthmatic HBE cells (blue symbols; **A**–**C**).

**Figure 3 cells-11-00256-f003:**
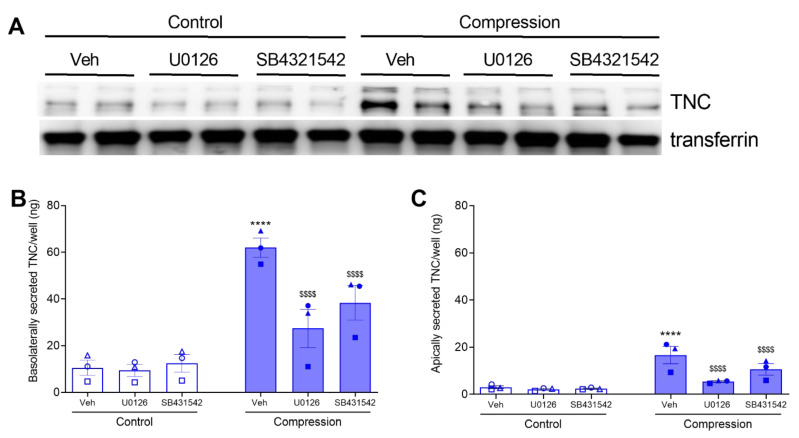
Compression-induced TNC secretion depends on TGF-β receptor and ERK pathways. TNC secreted by HBE cells at 24 h post-compression, in the presence or absence of U0126 or SB431542. Representative Western blot of three independent experiments shows the detection of basolateral secretion of TNC. Transferrin was detected as a loading control (**A**). Secreted TNC were detected by ELISA in the basolateral conditioned media (**B**) and in the apical washes (**C**) (mean ± SEM, 3 non-asthma donors). **** *p* < 0.0001, significantly different from vehicle control; ^$$$$^ *p* < 0.0001, significantly different from vehicle with pressure, analyzed by two-way ANOVA with Bonferroni’s post-hoc test. Each symbol represents each of the three donors. The open symbols represent control condition, and the closed symbols represent compressed condition in non-asthmatic HBE cells (blue symbols; **B**,**C**).

**Figure 4 cells-11-00256-f004:**
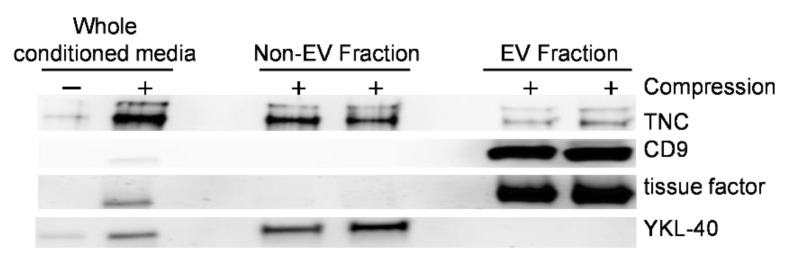
Mechanical compression induces the release of EVs containing TNC from HBE cells. Representative Western blots of four independent experiments show that CD9, tissue factor, YKL-40, and TNC were detected in the whole conditioned media with a greater abundance in conditioned media from compressed cells. YKL-40 was detected in the non-EV fraction, while CD9 and tissue factor were detected in the EV fraction. TNC was detected in both the non-EV and the EV fractions.

**Table 1 cells-11-00256-t001:** Primer sequences used in RT-qPCR [42].

Genes	Primer Sequences
*GAPDH*	FW: 5′-TGGGCTACACTGAGCACCAG-3′ RV: 5′-GGGTGTCGCTGTTGAAGTCA-3′
*TNC*	FW: 5′-TCCCAGTGTTCGGTGGATCT-3′ RV: 5′-TTGATGCGATGTGTGAAGACA-3′

## Data Availability

Data is contained within the article or Appendix A.

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
