# Peer review of "Mechanical Compression of Human Airway Epithelial Cells Induces Release of Extracellular Vesicles Containing Tenascin C"

_cells, 2022, doi:10.3390/cells11020256_

Round 1
Reviewer 1 Report
In their interesting study Mwase Ch et al. investigated the mechanisms of mechanical compression-induced expression of tenascin C (TNC) in differentiated human bronchial epithelial cells (HBEC) from normal donors and asthma patients (n=12 in total, however, there are only 3 to 6 subj. per experiment/group). This manuscript fits well into a broader area of research, including several papers from the same group, on the potential influence of mechanical stress associated with bronchospasm on the progression of airway remodeling. The authors used interesting cell culture model in which air pressure (30 cmH2O) is applied to air-liquid interface (ALI) differentiated HBEC. The most important result is confirmation that such compression activates bronchial epithelial cells in a manner similar to growth/profibrotic factors (TGF-b) and is dependent on the TGF-B receptor/ERK pathway. The authors attempted to explain whether TNC is secreted via exosomes, yet only one experiment performed shows predominantly non-exosome route with only small amount detected within exosome fraction (of basolateral medium). In my opinion, this issue is still open. The study has some clinical implications, since bronchoconstriction associated with the symptoms of obstructive airway diseases exerts a similar physical pressure (P) on the epithelium (~30 cmH2O). Interestingly, comparable air pressures are also used as a widely accepted threshold for plateau-P in assisted ventilation, suggesting a possible induction of the profibrotic phenotype also in other clinical settings/treatments. An interesting aspect, not addressed by this study, is whether compression itself could induce remodeling of bronchial epithelium (e.g., mesenchymal transition) and if such changes are long lasting, i.e., beyond periods of the P-stress associated with asthma attacks. The study is clear and concise, yet in its current form, a low number of replicates in some experiments and moderate novelty (only one protein studied, downstream pathways only partially explored, linkage with asthma phenotype not supported by solid data) weaken its overall merit. I don’t have many critical comments, though.
Major
- Interpretation of data related to the asthma phenotype. The study groups are very small, thus any conclusions on potential asthma-related phenotype (intrinsic, epigenetic?) of cells derived from asthma subjects should be restrained. For example, in Fig. 1 the only difference between dataset 3hr and 24hr (control condition, i.e., no pressure applied) was that they were derived from different wells, ~20 hours apart. As these were cells from a single donor, only collected some time later, the significance at 3hr (or lack thereof at 24hr) should be just accidental due to a natural variance in the epithelial growth in ALI. Additionally, asthma is very heterogeneous in terms of the type of airway inflammation, severity, pattern of ventilatory impairment, etc. Much larger groups are needed to draw reliable conclusions. Moreover, asthmatic cell lines were not further used in the inhibition experiment. To avoid misinterpretation, the Authors should consider combining control and asthma data in the first part of results (Chapter 3.1, Fig. 1) to reach higher number of replicates and statistical power of the crucial comparison (control vs. compression). I suggest cautiously interpreting the data on the differences between asthma vs. control, or just moving this to the supplement.
Minor
- RNA-seq data. The Authors used bulk RNA-seq to “identify mechanical compression-responsive genes that are regulated by either ERK or TGF-b…” (lines 210-211), yet these results are not presented. Next, they just show data on the influence of ERK or TGF-b receptor inhibition on TNC expression. I assume, this part of the data refers to their earlier seq study (Ref. 42), which used an identical setup (3 and 24 hrs, pressures, etc.), showed marked up-regulation of TNC gene mRNA, ERK pathway stimulation, and TGF-b signaling processes after compression. The authors have already checked for TNC expression in the first place (Fig. 1, they know what to look for), so there is no need to do whole RNA sequencing to confirm that ERK inhibition decreases TNC mRNA. To make this point clearer, the Authors should present and discuss seq data, or at least explain what was the reason for RNA-seq in the context of the presented data.
- Please check Figure legends (e.g., D in Fig.1; CD in Fig.3).
- Basolateral secretion of TNC is not surprising as TNC mRNA is expressed almost exclusively by basal cells of airway epithelium, at least according to single cell seq studies (e.g., lung cell atlas at https://cells.ucsc.edu). The potential spatial distribution and cell type dependence should be discussed.
- It seems like some experiments were performed only in one replicate (e.g., Fig. 3a, Fig. 4), this should be pointed in the methods section or Fig. legends. If not, please provide relevant data on the number of replicates, densitometry, and statistics. I assume that, in the case of TNC detection in cell culture supernatants, WB was only a complementary method to confirm ELISA results.
- The exact mechanism of TNC hexabrachions secretion in airway epithelial cells has not been studied in detail, however recent papers mostly dealing with mesenchymal or cancer cells (e.g., Ref 75, and reviewed in Ref 43) suggest exosome pathway. The last experiment presented in the manuscript (chapter 3.3) suggest rather non-exosome pathway. On the basis of this single WB it is hard to judge how significant is the contribution of exosomes in this process. Nevertheless, the exosome experiment announces interesting results and should be repeated (replicates needed) to yield more solid data. Why not solve this issue, e.g., by using specific inhibitors of the exosome or MV pathway.
Author Response
We thank the reviewer for taking the time to review our manuscript and providing insightful comments on the submitted manuscript. Attached please see the responses.

Reviewer 2 Report
The manuscript is well-written, novel, and very interesting. I read it with high interest and pleasure.
It examines compression-induced expression and secretion of the extracellular matrix protein tenascin C (TNC) from well-differentiated primary human bronchial epithelial cells obtained from asthma and control subjects and grown in ALI culture.
The authors indicated that at baseline, TNC mRNA expression was elevated, and secreted TNC protein was significantly higher in asthmatic compared to non-asthmatic cells. In response to mechanical compression, both TNC mRNA expression and secreted TNC were increased in both non-asthmatic and asthmatic cells, which was depended on both the ERK and TGF-β receptor pathways. Furthermore, mechanically compressed HBE cells released extracellular vesicles containing TNC in response to mechanical stress.
I have only two less important comments:
1. The three asthmatics were with fatal asthma, thus the observed phenomenon might be related only to the most severe form of the disease or brittle fatal asthma. Could the authors provide more detail on the clinic of asthma patients: spirometry changes before fatality? Asthma duration? Reason of death? Patients' age? The eosinophilic or not-eosinophilic phenotype? Atopy?, etc. The same question about the fourth asthmatic: what kind of asthma was that: mild, moderate? In the case of the less severe form of the disease, the results would be even more exciting, pointing to the increased risk of airway remodeling in all asthma subjects. Please provide also a short comment on that issue in the Discussion section.
Clinical data of asthma patients could be in a Supplementary file, as stated in the text (Figure S1), however, the Supplement was not attached to my reviewer account.
2) Please check the Figure 1 description: missed „D”. Furthermore it is slightly challenging to follow this Figure. The authors could improve the description to clarify results, e.g. A panel: TNC mRNA expression (Y-axis description) please also add „uncompressed cells” or „at baseline”- it is not clear on first sign.
Author Response

(The authors gave the same response as above.)
